# Proposed therapy, developed in a *Pcdh15*-deficient mouse, for progressive loss of vision in human Usher syndrome

**Saumil Sethna[1], Wadih M Zein[2], Sehar Riaz[1,3], Arnaud PJ Giese[1], Julie M Schultz[4†], Todd Duncan[5], Robert B Hufnagel[2], Carmen C Brewer[6], Andrew J Griffith[6‡], T Michael Redmond[5], Saima Riazuddin[1], Thomas B Friedman[4], Zubair M Ahmed[1,7,8]\***

[1]Department of Otorhinolaryngology - Head & Neck Surgery, University of Maryland School of Medicine, Baltimore, United States; [2]Ophthalmic Genetics and Visual Function Branch, National Eye Institute, National Institutes of Health, Bethesda, United States; [3]National Center of Excellence in Molecular Biology, University of the Punjab, Lahore, Pakistan; [4]Laboratory of Molecular Genetics, National Institute of Deafness and Other Communication Disorders, National Institutes of Health, Bethesda, United States; [5]Laboratory of Retinal Cell and Molecular Biology, National Eye Institute, National Institutes of Health, Bethesda, United States; [6]Otolaryngology Branch, National Institute of Deafness and Other Communication Disorders, Bethesda, United States; [7]Departments of Ophthalmology and Visual Sciences, University of Maryland School of Medicine, Baltimore, United States; [8]Departments of Molecular Biology and Biochemistry, University of Maryland School of Medicine, Baltimore, United States

**\*For correspondence:**
zmahmed@som.umaryland.edu

**Present address:** †GeneDx, Inc, Gaithersburg, United States; ‡Departments of Otolaryngology and Physiology, College of Medicine, University of Tennessee Health Science Center, Memphis, United States

**Abstract** Usher syndrome type I (USH1) is characterized by deafness, vestibular areflexia, and progressive retinal degeneration. The protein-truncating p.Arg245\* founder variant of *PCDH15* (USH1F) has an ~2% carrier frequency amongst Ashkenazi Jews accounts for ~60% of their USH1 cases. Here, longitudinal phenotyping in 13 USH1F individuals revealed progressive retinal degeneration, leading to severe vision loss with macular atrophy by the sixth decade. Half of the affected individuals were legally blind by their mid-50s. The mouse *Pcdh15*[R250X] variant is equivalent to human p.Arg245\*. Homozygous *Pcdh15*[R250X] mice also have visual deficits and aberrant light-dependent translocation of the phototransduction cascade proteins, arrestin, and transducin. Retinal pigment epithelium (RPE)-specific retinoid cycle proteins, RPE65 and CRALBP, were also reduced in *Pcdh15*[R250X] mice, indicating a dual role for protocadherin-15 in photoreceptors and RPE. Exogenous 9-*cis* retinal improved ERG amplitudes in *Pcdh15*[R250X] mice, suggesting a basis for a clinical trial of FDA-approved retinoids to preserve vision in USH1F patients.

## Introduction

Usher syndrome (USH) is estimated to be responsible for more than 50% of deaf-blind cases, 8–33% of patients with retinitis pigmentosa (RP), and 3–6% of congenitally deaf individuals (*Boughman et al., 1983*; *Brownstein et al., 2004*; *Vernon, 1969*). Clinical data review studies estimated a prevalence of 3.2–6.2 per 100,000 for USH cases (*Boughman and Fishman, 1983*; *Koenekoop et al., 1993*). However, a molecular diagnosis study in children with hearing loss found variants in USH-associated genes in 11 % and estimated a frequency of 1/6000 individuals afflicted with USH in the United States

(*Kimberling et al., 2010*). Assuming similar prevalence, this would translate into 255,000–1.34 million USH cases worldwide. However, this estimate varies considerably in specific population substructures. For instance, the p.Arg245* founder variant of *PCDH15* (USH1F) has an ~2% carrier frequency amongst Ashkenazi Jews accounts for nearly 60% of their Usher syndrome type I (USH1) cases (*Ben-Yosef et al., 2003*). Thus, we speculate that a comprehensive understanding of the pathophysiology and disease mechanisms is a prelude for developing therapeutic interventions for USH after clinical trials.

Loss of vision in individuals with USH1, an autosomal recessive disorder, begins towards the end of their first decade of life due to RP, eventually leading to near total blindness. Night blindness is an early sign in USH1 subjects followed by constriction of the visual field (tunnel vision) and finally clinical blindness (*Vernon, 1969*). Characteristic fundus features include pigmentary retinopathy, narrowing of the retinal vessels, and a pale appearance of the optic disk (*Toms et al., 2020*). Vestibular dysfunction in USH1 manifests as a delay in development of independent ambulation while hearing loss is usually severe to profound, congenital, and sensorineural (*Ahmed et al., 2003b*; *Smith et al., 1994*). Cochlear implants can restore auditory perception in USH1 patients (*Brownstein et al., 2004*; *Pennings et al., 2006*), but presently there is no effective treatment for vision loss due to RP. Moreover, there is a lack of longitudinal data for the natural history of ocular abnormalities associated with variants of *PCDH15* in humans. Only anecdotal clinical data has been reported thus far (*Ahmed et al., 2001*; *Ben-Yosef et al., 2003*; *Brownstein et al., 2004*; *Jacobson et al., 2008*). Here, we describe the natural history of retinopathy in 13 individuals followed for up to 30 years with an Ashkenazi Jewish recessive founder variant of PCDH15. Eleven patients from nine families were homozygous for the p.Arg245*, leading to truncation of the encoded protein, protocadherin-15. Two additional patients had compound heterozygous genotypes that included one p.Arg245* allele.

Protocadherin-15 is a member of a large cadherin superfamily of calcium-dependent cell–cell adhesion molecules (*Ahmed et al., 2001*; *van Roy, 2014*). Within the vertebrate inner ear, protocadherin-15 is required for the structural maintenance and the mechanotransduction function of the sensory hair cells (*Ahmed et al., 2006*; *Kazmierczak et al., 2007*). In the retina, protocadherin-15 is localized to the outer limiting membrane of photoreceptors (PRs) and in Müller glia (*Reiners et al., 2005b*; *van Wijk et al., 2006*). We previously reported a reduction of electroretinogram (ERG) a- and b-waves amplitudes (~40%) at 5 weeks of age in at least two *Pcdh15* alleles in mice (*Pcdh15*[av-5J] and *Pcdh15*[av-jfb]) (*Haywood-Watson et al., 2006*). However, the exact molecular function of protocadherin-15 in the retina remains elusive. Here, we describe the pathophysiology and function of protocadherin-15 in the retina of a novel murine model. Finally, ERG data show significant improvement after treatment of this USH mouse model with 9-*cis* retinal, raising the possibility that exogenous retinoids could preserve vision in USH1F patients.

## Results and discussion

### Spectrum and longitudinal ocular phenotypic data revealed early onset rod-cone dystrophy in USH1 subjects

We reviewed the medical records of 13 patients enrolled in an Institutional Review Board-approved protocol to study USH. Subsequent to congenital profound deafness, the first reported symptom was difficulty with vision at night. Ophthalmic manifestations depended on the age of the patient and the stage of retinal degeneration (*Table 1*). Electroretinography (ERG) recordings were at noise-level for both scotopic and photopic responses, suggesting dysfunctional PRs (*Figure 1—figure supplement 1*). In young patients with early stages of the retinal degeneration, findings included mottling of the retinal pigment epithelium (RPE) with early pigment redistribution, and mild to moderate vascular attenuation with typically preserved macular reflexes (*Figure 1a*). As RP progressed, more extensive pigment abnormalities were observed with deposition of bone spicules, severe attenuation of the retinal vasculature, macular atrophic changes, and waxy pallor of the optic nerve head (*Figure 1b*). In advanced stages, these changes became more prominent and widely distributed throughout the fundus (*Figure 1c*, *Figure 1—figure supplement 1*). Cataracts were common, especially posterior subcapsular opacities. The panels in *Figure 1d* show the progression of macular atrophic changes over a 12-year period in a patient with compound heterozygous p.Arg245*/p.Arg929* variants. Both of these alleles of *PCDH15* are predicted to cause truncation of the protocadherin-15 protein.

**Table 1.** Ophthalmic clinical manifestations of patients with biallelic *PCDH15* mutations.
Eleven patients in this study are homozygous for the p.Arg245* variants while two siblings carry compound heterozygous variants, ¥p.Arg245*/p.Arg929*. Visual acuity assessments consistently show a decline between the third and fourth decade of life. Visual field loss in these subjects, shown in Table 1, further support severe retinitis pigmentosa. Macular atrophy and PSC cataract appear early and may contribute to the observed reduction in visual acuity. Optic nerve head pallor is also frequent and advanced. Normal Goldmann Visual Field Perimetry horizontal diameters are in the range of: V4e—150–160° , I4e—130–140°, and I1e—20–30°. BCVA: Best-Corrected Visual Acuity, OD: right eye, OS: left eye, HM: Hand Motion visual acuity, LP: Light Perception, HVF: Humphrey Visual Field, PSC: Posterior Subcapsular Cataract, NS: Nuclear Sclerosis Cataract, IOL: Intraocular lens, CME: Cystoid Macular Edema, ERM: Epiretinal Membrane, RPE: Retinal Pigment Epithelium, ONH: Optic Nerve Head, ND: Not Done, N/A: Not Available.

| | Age (years) | BCVA OD; OS | Visual field OD | Visual field OS | Lens | Macula | Spicules/ mottling | Optic nerve |
|---|---|---|---|---|---|---|---|---|
| LMG210 #1563 | 12 | 20/25; 20/30 | 90, ND, ND | 90, ND, ND | Clear | Normal | N/A | N/A |
| | 22 | 20/40; 20/50 | 30, 0, 0 | 28, 0, 0 | Mild PSC | Pigment | Spicules | Pale |
| | 35 | HM; HM | 20, 0, 0 | 15, 0, 0 | PSC | Atrophy | Spicules | Pale +3 |
| LMG279 #1795 | 19 | 20/25; 20/32 | 50, 15, 0 | 65, 18, 0 | PSC | Normal | Spicules | Pale |
| LMG268 #1722 | 37 | 20/30; 20/30 | 23, 14, 10 | 20, 12, 10 | PSC | Normal | Spicules | Pale |
| | 50 | 20/160; 20/50 | 12, 1, 0 | 15, 2, 0 | PSC | Atrophy OD | Spicules | Pale |
| | 59 | LP; 20/500 | 0, 0, 0 | 6, 0, 0 | PSC+NS | Atrophy | Spicules | Pale |
| | 67 | LP; HM | ND | ND | PSC+NS OD; IOL OS | Atrophy | Spicules | Pale |
| LMG178 #1463 | 25 | 20/100; 20/30 | ND | ND | Mild PSC | CME | Spicules | ONH swelling |
| | 26 | 20/50; 20/40 | 30, 8, 0 | 35, 25, 0 | Mild PSC | CME | Spicules | Resolved swelling |
| LMG200 #1539 | 8 | 20/30; 20/30 | ND | ND | N/A | N/A | N/A | N/A |
| | 25 | 20/50; 20/40 | 40, 0, 0 | 40, 0, 0 | Clear | Atrophy | Spicules | Pale+1 |
| LMG200 #1538 | 6 | 20/40; 20/30 | ND | ND | N/A | N/A | N/A | N/A |
| | 22 | 20/100; 20/60 | 6, 0, 0 | 9, 0, 0 | Clear | Atrophy | Spicules | Pale+1 |
| LMG186 #1484 | 21 | 20/30; 20/40 | HVF Diameter 5° | HVF Diameter 10° | Mild PSC | Normal | Spicules | Pale |
| LMG407 #2149 | 55 | 20/400; 20/300 | 3, 0, 0 | 5, 0, 0 | IOL OU | Atrophy | Spicules | Pale +3 |
| LMG322 #1917 | 22 | 20/60; 20/30 | 140, 30, 0 | 145, 30, 0 | Clear | CME | Spicules | Normal |
| LMG322 #1916 | 11 | 20/25; 20/25 | 110, 35, 0 | 110, 30, 0 | Clear | Normal | Spicules | Normal |

*Table 1 continued on next page*

Table 1 continued

| | Age (years) | BCVA OD; OS | Visual field OD | Visual field OS | Lens | Macula | Spicules/ mottling | Optic nerve |
|---|---|---|---|---|---|---|---|---|
| | 16 | 20/25; 20/25 | 110, 35, 0 | 110, 30, 0 | Clear | Normal | Spicules | Normal |
| LMG125 #1221 | 12 | 20/30; 20/30 | ND | ND | Clear | CME, ERM | Spicules | Pale |
| LMG197 #1831¥ | 30 | 20/50, 20/50 | 80, 7, 0 | 100, 10, 0 | PSC | Normal | Spicules | Pale +1 |
| | 38 | 20/60, 20/60 | 40, 0, 0 | 60, 4, 0 | PSC | Atrophy | Spicules | Pale +2 |
| | 50 | 20/160, 20/100 | 15, 0, 0 | 22, 0, 0 | PSC | Atrophy | Spicules | Pale +3 |
| LMG197 #1839¥ | 27 | 20/60, 20/125 | 140, 15, 0 | 140, 15, 0 | Clear | ERM | RPE atrophy | Pale +1 |
| | 37 | 20/125, 20/250 | 65, 13, 0 | 55, 10, 0 | Clear | ERM | Spicules | Pale +3 |
| | 52 | 20/250, 20/250 | 35, 12, 0 | 30, 4, 0 | PSC +1 | Atrophy | Spicules | Pale +3 |

Kinetic visual field testing (*Table 1*) showed early loss of the ability to detect the smaller and dimmer target (I1e) in all but one patient (LMG268 #1722 at age 37 years). Early midperipheral scotomas and severe constriction were noted while testing the I4e isopter (target is equal in size to I1e but brighter). Progressive constriction of the V4e visual field isopter (largest and brightest target) is seen in *Figure 1e* where the horizontal diameter is binned by the decade of life. *Figure 1f* shows an increase in logMAR visual acuity, corresponding to a decline in Snellen best-corrected visual acuity (BCVA), starting at the fourth decade of life. Kaplan-Meier survival curves (*Figure 1g*) with parameters corresponding to legal blindness (acuity at 20/200 and visual field limited to 20° ) demonstrate severe visual function loss by the fifth decade.

## *Pcdh15^{R250X}* knockin mutant recapitulates human p.Arg245* Usher phenotype

In order to investigate the precise role of protocadherin-15 in light transduction and the mechanism of visual deficits observed in patients homozygous or compound heterozygous for the recessive Arg245* pathogenic variant of *PCDH15*, we used CRISPR/Cas9 technology to engineer a mouse model with the *Pcdh15^{R250X}* variant (Materials and methods) (*Cong et al., 2013*). Immunostaining showed that protocadherin-15 is localized to the inner segments of the PR, the outer plexiform layer, and the ganglion cell layer as reported previously (*Garwin et al., 2000*). However, the RPE was not assessed for the localization of protocadherin in those studies. Using immunohistochemistry, here we show that protocadherin-15 is also expressed in the RPE (*Figure 2—figure supplement 1*). The p.Arg250* variant is located in exon 9 common to all *Pcdh15* transcripts and, consequently, is predicted to cause a complete loss of all known protocadherin-15 isoforms (*Ahmed et al., 2006*). Indeed, with immunostaining, we could not detect protocadherin-15 expression in retinal tissue (*Figure 2—figure supplement 1a-b*) or cochlear tissue (*Figure 2—figure supplement 1c*) from *Pcdh15^{R250X/R250X}* mutant mice.

Consistent with previously published *Pcdh15* mouse models (*Alagramam et al., 2011*; *Garwin et al., 2000*; *Senften et al., 2006*), we detected no auditory-evoked brainstem responses (ABRs) in mutant *Pcdh15^{R250X}* mice at P16, the earliest postnatal day that ABRs can be reliably detected (*Figure 2—figure supplement 2a*), indicating that they were profoundly deaf. Furthermore, *Pcdh15^{R250X}* mutant mice displayed abnormal motor vestibular behaviors such as circling, hyperactivity, and head bobbing. Behavioral tests including exploratory behavior and tail-hanging tests confirmed that these deaf mice also have a significant vestibular dysfunction (*Figure 2—figure supplement 2b, c*). Finally the *Pcdh15^{R250X}* mutant cochlear and vestibular hair cells also had no functional mechano-transduction (*Figure 2—figure supplement 2d*), accounting for deafness, and at P60 also showed

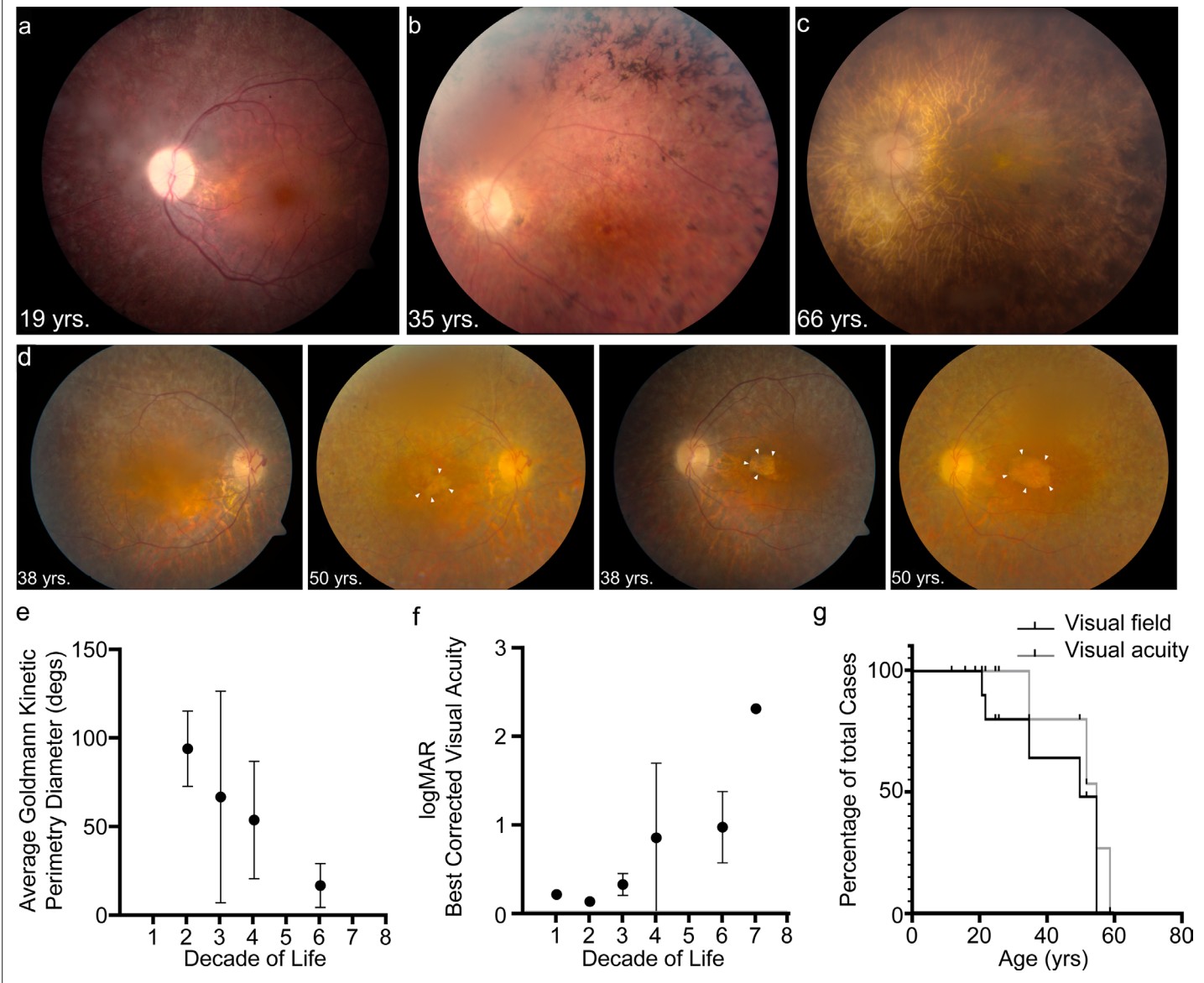

**Figure 1.** USH1F p.Arg245* spectrum and longitudinal eye phenotype. (**a–c**) Fundus images depicting the spectrum of retinal findings in p.Arg245* USH1F patients, show mottling of pigment epithelium, attenuation of retinal vasculature, and pallor of optic nerve head seen in all three fundus photos. Peripheral bony spicules and macular atrophy are noted (**b**). Diffuse atrophy and advanced retinal degeneration are seen in (**c**). (**d**) Longitudinal progression of macular atrophic changes (arrowheads point at edge of macular atrophic area) over a 12-year period in a USH1F patient who is compound heterozygous for p.Arg245*/p.Arg929*. (**e**) Mean and SEM of Goldmann visual field diameters for patients with data binned by decade of life. (**f**) Mean and SEM of best-corrected visual acuity binned by decade of life for all patient visits. (**g**) Survival analysis curves for visual acuity (logMAR visual acuity>1 , i.e., acuity worse than 20/200) and visual field (visual field <20° in better eye). These values were chosen since they usually denote visual function at legal blindness levels. SEM, standard error of the mean. The online version of this article includes source data and the following figure supplement(s) for Figure 1.

The online version of this article includes the following figure supplement(s) for figure 1:

**Figure supplement 1.** Fundus autofluorescence, optical coherence tomography (OCT), and electroretinography in compound heterozygous patient LMG197 #1831.

degeneration of hair cells in the organ of Corti (**Figure 2—figure supplement 3**). Taken together, our data indicate that the *Pcdh15*[R250X] mutants recapitulate human p.Arg245* deafness and peripheral vestibular areflexia.

To parallel the visual examinations performed in patients, we assessed the visual function of *Pcdh15*[R250X] mutant mice using full-field ERG. Dark-adapted (scotopic) ERG waves, which are

preferentially driven by rod PRs at low light intensity and by rod and cone PRs at high light intensity, showed normal wave architecture albeit with reduced amplitudes (*Figure 2—figure supplement 4a*). Quantification showed significant reduction in amplitudes of the a-wave derived primarily from the PR layer and the b-wave derived from Müller glia and bipolar neurons in 1-month-old *Pcdh15^{R250X}* mutant mice as compared to littermate control mice (*Figure 2a*). Similarly, photopic ERG amplitudes, primarily representing cone-mediated function, were also reduced in 1-month-old *Pcdh15^{R250X}* mutant mice (*Figure 2b*). The b- to a-wave ratio (b/a) was similar across genotypes indicating that deficits were manifested mainly at the PR level (*Figure 2—figure supplement 4b*). We then performed ERGs at 2–3, 6–7, and 12–14 months of age. *Pcdh15^{R250X}* mutant mice consistently had lower scotopic and photopic ERG amplitudes compared to controls (*Figure 2c–h*), indicating that the functional deficits observed at 1 month were not due to delayed development. Further, to assess the visual cycle dysfunction and dark adaptation, we assessed the recovery of a-wave amplitude following bleaching of more than 90% of rhodopsin (*Kolesnikov et al., 2020*). We observed equivalent recovery of a-wave irrespective of genotype (*Figure 2—figure supplement 4c*), but we did observe that initial single flash a-wave amplitude was lower in mutant mice (*Figure 2—figure supplement 4d*), which suggests functional alterations in phototransduction such as are indicated by aberrant light-dependent translocation of arrestin and transducin (see details below). To correlate functional deficits with the structural integrity of the retina, we performed non-invasive in vivo retinal imaging using optical coherence tomography (OCT) in young (1–2 months) and old (12–14 months) mice, which showed no gross structural abnormality in homozygous *Pcdh15^{R250X}* mutants (*Figure 2i*). However, we did note a small but significant decrease in the outer nuclear layer (ONL) thickness in 12–14 months old mutant mice (*Figure 2j*).

## Mechanisms contributing to ERG defects due to protocadherin absence

We hypothesized that the functional deficits, reflected by abnormal ERG findings, without structural impairment of the retina might result from deficits in the phototransduction cascade or the visual retinoid cycle. The phototransduction cascade mediates the transduction of light into neuronal signals, while the visual retinoid cycle regenerates a key chromophore, 11-*cis* retinal. The rod outer segments (OS) are exquisitely adapted for light transduction. Phototransduction proteins are generated in the PR cell body and delivered to the OS via the inner segment (IS) and connecting cilium. Under photopic conditions (daylight), arrestin translocates from IS of the PRs to the OS, to desensitize the opsin. Conversely, transducin translocates from OS to IS of the photoreceptors allowing arrestin to bind to opsin (*Arshavsky et al., 2002*; *Burns and Baylor, 2001*). We found significant mislocalization of both arrestin and transducin to the PR IS and OS in *Pcdh15^{R250X}* mutant mice under light-adapted conditions, whereas transducin was correctly localized only to the IS and arrestin only to the OS in control mouse retinae (*Figure 3a*, *Figure 3—figure supplement 1*). In dark-adapted conditions, arrestin was correctly localized to the IS and transducin to the OS in both mutant and control mice (*Figure 3—figure supplement 1a*, c-d). Finally, opsin was correctly localized only in the OS under both dark- and light-adapted conditions (*Figure 3—figure supplement 1b*, e), indicating that protocadherin-15 is essential for rapid shuttling of proteins from IS to OS and vice-versa in response to adaptation to light.

The 11-*cis* retinal complexes with opsin to form rhodopsin. Absorption of a single photon by 11-*cis* retinal leads to its photo-isomerization to all-*trans*-retinal within femtoseconds (*Nogly et al., 2018*), thus activating opsin and initiating the phototransduction cascade. Consequently, there is decoupling of opsin and all-*trans*-retinal. All-*trans*-retinal must be re-isomerized to 11-*cis* retinal to form rhodopsin again. These enzymatic steps occur in the RPE (*Saari, 2000*; *Travis et al., 2007*; *Wald and Brown, 1956*). Next, we assessed the levels of crucial retinoid cycle proteins such as RPE65, an essential isomerase that catalyzes the conversion of all-*trans*-retinyl ester to 11-*cis* retinol (*Jin et al., 2005*; *Moiseyev et al., 2005*; *Redmond et al., 2005*), CRALBP (cellular retinaldehyde-binding protein), a key retinoid transporter, and IRBP (interphotoreceptor retinoid-binding protein). These studies were rationalized based on the findings that protocadherin-15 is a binding partner of myosin VIIA (*Senften et al., 2006*), which also interacts with RPE65 (*Lopes et al., 2011*). Similar to protocadherin-15, pathogenic variants of myosin VIIA also causes USH1 (*Jacobson et al., 2011*; *Weston et al., 1996*). Intriguingly, immunoblotting revealed significantly reduced quantities of RPE65 and CRALBP, but not IRBP in *Pcdh15^{R250X}* mutant mice (*Figure 3b and c*).

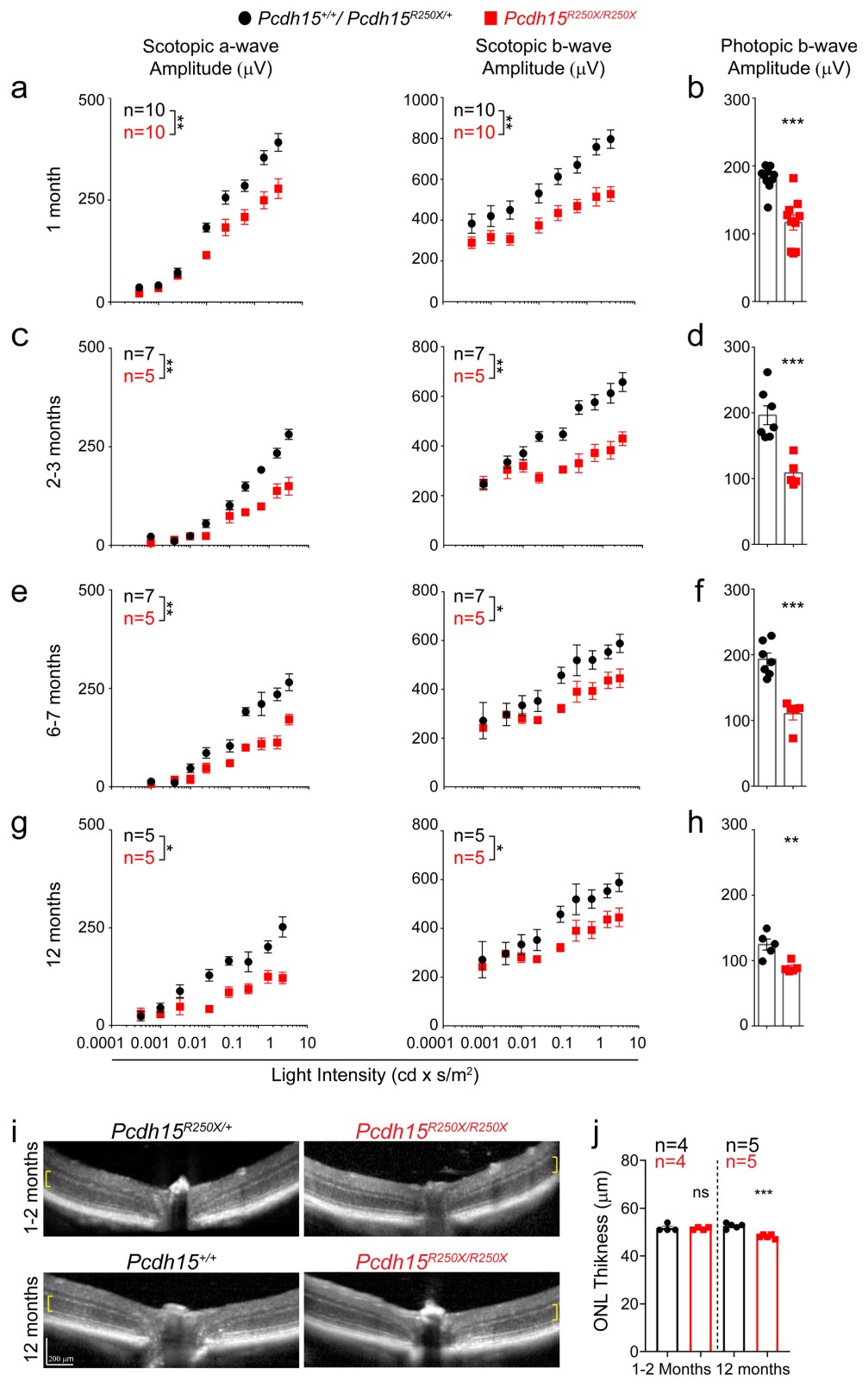

**Figure 2.** Loss of protocadherin-15 leads to visual dysfunction over a period of 1 year. (**a**) Quantification of scotopic (dark adapted) responses from littermate control (Pcdh15^+/+ or Pcdh15^R250X/+) and mutant (Pcdh15^R250X/R250X) mice at 1 month of age revealed progressive loss of both a- (*left* panels) and b-wave (*right* panels) amplitudes in mutant mice. Representative ERG waveforms are shown in Extended data *Figure 3b*. (**b**) Quantification of photopic (light

*Figure 2 continued on next page*

*Figure 2 continued*

adapted) b-wave indicates decline of cone photoreceptor function in mutant mice. (**c–h**) Quantification of scotopic ERG amplitudes (**c, e, g**) and photopic ERG amplitudes (**d, f, h**) at indicated ages shows sustained decline in amplitudes over time in *Pcdh15* mutant mice. (**i**) Representative OCT images from mice of denoted genotype, shows no gross retinal degeneration in young (1–2 months, top panels) or old (12–14 months, bottom panels) mice. (**j**) Quantification of outer nuclear layer (ONL) of images showed in (**i**), shows mild loss of ONL in aged mutant mice. Data presented as mean ± SEM. Each data point represents an individual mouse. Data presented as mean ± SEM. Student's unpaired t-test, p<0.05 (*), p<0.01 (**). The online version of this article includes source data and the following figure supplement(s) for Figure 2.

The online version of this article includes the following figure supplement(s) for figure 2:

**Figure supplement 1.** Validation of loss of protocadherin-15 in the retina and cochlea of *Pcdh15$^{R250X}$* mutant mice.

**Figure supplement 2.** *Pcdh15$^{R250X}$* mutant mice have profound hearing loss and severe vestibular system dysfunction.

**Figure supplement 3.** *Pcdh15$^{R250X}$* mutant mice have degeneration of sensory hair cells in the organ of Corti (**a, b**).

**Figure supplement 4.** Loss of protocadherin-15 leads to retinal dysfunction in *Pcdh15$^{R250X}$* mutant mice.

We also quantified the absolute retinoid levels within the eyes after overnight dark adaptation, and as compared to controls found reduced levels of retinoids, particularly 11-*cis*-retinaloxime, in *Pcdh15$^{R250X}$* mutant mice (*Figure 3d*). 11-*cis* retinaloxime levels correlate with PR rhodopsin levels. Next, we quantified the retinoid levels 1 hr after dark adaptation following bleaching with 15,000 lux for 1 hr (*Li et al., 2019*). Reduction of 11-*cis* retinaloxime in control and mutant mice retinae (*Figure 3d*) correlated with prebleach levels, as did increase in all-*trans*-retinyl esters. These findings from *Pcdh15$^{R250X}$* mutants suggest a reduced function of the visual cycle due to reduced expression of RPE65 and CRALBP. Since we observed lower visual cycle proteins (RPE65 and CRALBP), we also assessed the structure of the RPE, the main cell type harboring the key enzymes of the visual cycle. Transmission electron microscopy showed no gross structural deficits in the RPE (*Figure 3—figure supplement 2*). Taken together, our data indicate that the loss of protocadherin-15 in the retina leads to aberrant translocation of proteins involved in the phototransduction cascade and reduced levels of key retinoids and enzymes involved in the visual retinoid cycle.

## Pre-clinical administration of exogenous retinoids

We hypothesized that low levels of retinoids in the mutant mice could be overcome by providing exogenous retinoids, thus rescuing the ERG deficits (*Palczewski, 2010*; *Sethna et al., 2020*). To test this hypothesis, we first performed baseline ERGs on 2–3-month-old control and *Pcdh15$^{R250X}$* mutant mice. After 1 week, *Pcdh15$^{R250X}$* mutant mice were injected intraperitoneally (IP) with 9-*cis* retinal, an analog of naturally occurring 11-*cis* retinal. Control mice were injected with vehicle. ERGs were performed the next day after overnight dark adaptation. Remarkably, a single treatment of *Pcdh15$^{R250X}$* mutant mice with 9-*cis* retinal was sufficient to increase their ERG amplitudes to levels comparable to those in vehicle-injected wild-type (WT) controls (*Figure 4a–b*, *Figure 4—figure supplement 1*). Similarly, we also observed an improvement in cone function to levels of the vehicle-injected control mice (*Figure 4d*). The b- to a-wave ratio was consistent with vehicle-injected control mice or baseline *Pcdh15$^{R250X}$* mutant mice (*Figure 4c*), suggesting a proportional increase in PR function after retinoid therapy. To confirm the 9-cis retinal delivery and metabolism, in a separate cohort of mice, 24 hr post-injection followed by 2 hr light exposure, we evaluated the retinaloxime levels, including 9-*cis* retinal-oxime, both in the liver and retina. As expected, we found trace levels of 9-*cis* retinaloxime levels in the liver and retinae (*Figure 4—figure supplement 1*).

Next, to assess the impact of exogenous retinoids in aged animals, we performed similar experiments using 6–7-month-old mice. We found a comparable increase in functional activity with a single IP injection of 9-*cis* retinal in mutant mice as compared to the same cohort of mutant mice assessed one week earlier (baseline *Pcdh15$^{R250X}$* mutant mice). The ERG amplitudes of 9-*cis* retinal-injected mutant mice were nearly indistinguishable from those of vehicle-injected WT control mice (*Figure 4d–f*). Finally, in a separate cohort of 6–7-month-old mutant mice, we assessed the longevity of the retinoid-mediated improvement. We found that by 2 weeks after 9-*cis* retinal treatment, the impact of exogenous retinoid treatment on ERG improvements was reduced (*Figure 4g*).

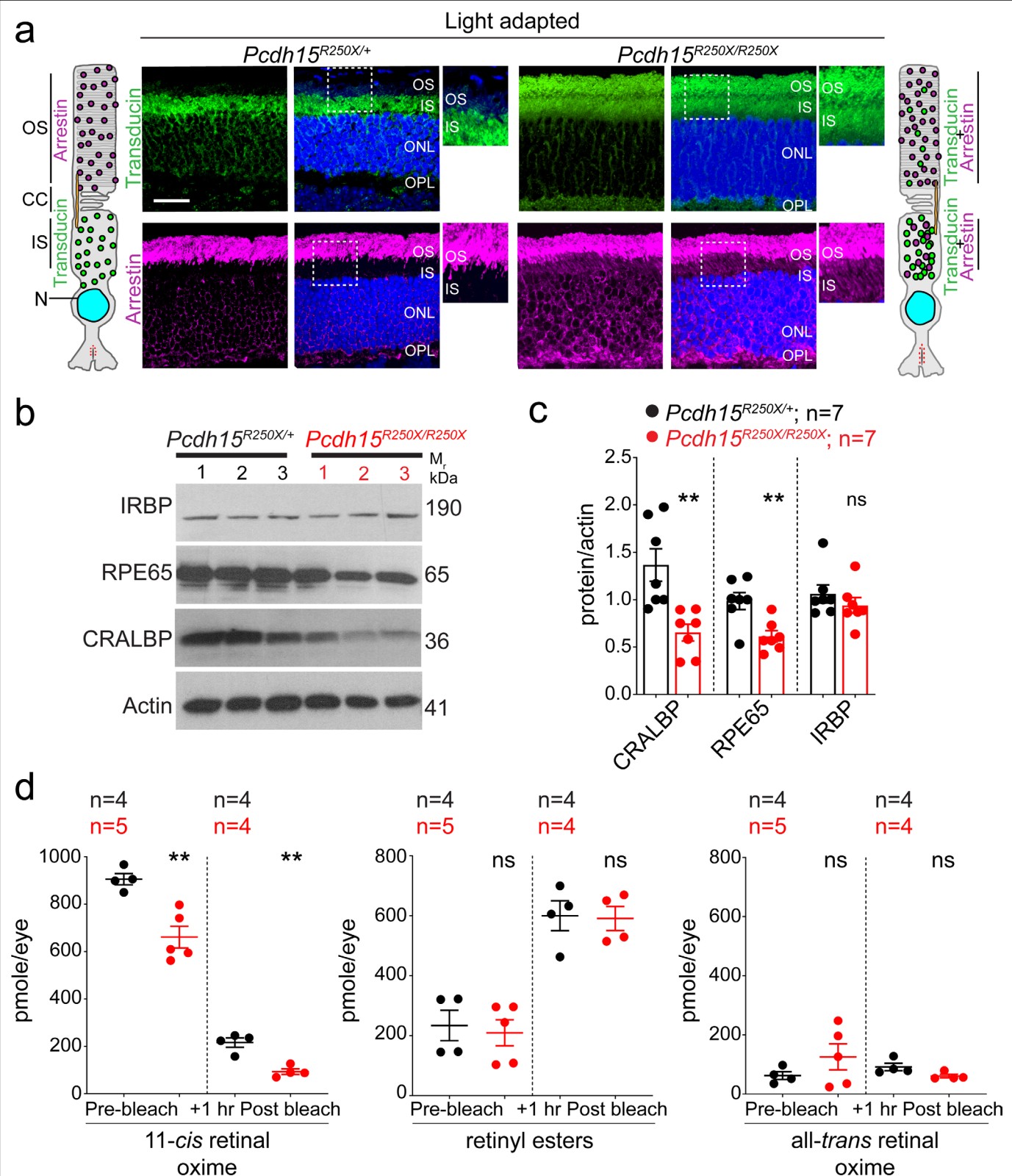

**Figure 3.** Loss of protocadherin-15 leads to aberrant localization of key proteins involved in the phototransduction cascade and retinoid cycle. (**a**) Representative confocal micrographs of light-adapted retinae show mislocalization of phototransduction cascade proteins, arrestin, and transducin, to both the inner segment (IS) and outer segment (OS) in mutant mice (*right* panels). In control mice, transducin is correctly localized to the IS and arrestin is to the OS (*left* panels). A schematic of the localization of arrestin and transducin in control and mutant mice is also shown. Scale bar: 20 μm.

*Figure 3 continued on next page*

*Figure 3 continued*

(**b, c**) Immunoblot of proteins involved in the visual retinoid cycle shows reduced quantities of RPE65 and CRALBP but not IRBP, quantified in (**c**). (**d**) Quantification of indicated retinoid species from control and mutant mice shows reduced quantities of 11-*cis* retinal oxime. Data presented as mean ± SEM. Each data point represents an individual mouse. Student's unpaired t-test, p<0.05 (*), p<0.01 (**). The online version of this article includes source data and the following figure supplement(s) for Figure 3. ONL, outer nuclear layer; OPL, outer plexiform layer.

The online version of this article includes the following figure supplement(s) for figure 3:

**Figure supplement 1.** Loss of protocadherin-15 does not affect dark-adapted localization of key phototransduction cascade proteins.

**Figure supplement 2.** Loss of protocadherin-15 did not apparent degeneration of RPE in *Pcdh15^{R250X/R250X}* mutant mice.

Next, we assessed whether exogenous 9-*cis* retinal treatment also improved translocation of arrestin and transducin in *Pcdh15^{R250X}* mutant mice. For these studies, we injected a cohort of mutant mice with either 9-*cis* retinal or vehicle. Overnight dark-adapted mice were exposed to normal room light for 2 hr and their retinae was examined for localization of arrestin and transducin. We found mislocalization of arrestin and transducin in mutant mice injected with 9-*cis* retinal as well (***Figure 4— figure supplement 2***). Taken together, our data link the visual deficits to the retinoid cycle dysfunction in *Pcdh15^{R250X}* mutant mice and provides a starting point to investigate the possibility of therapeutically boosting visual function in USH1F patients.

The spectrum and longitudinal ophthalmic phenotypes of USH1F patients homozygous for the p.Arg245* variant (or compound heterozygotes) consisted of a rod-cone dystrophy and are relatively uniform across this cohort of patients. They include an onset of symptoms such as night vision difficulties and visual field deficits in the first or early second decade, the presence of macular atrophy with reduction in central visual acuity by the third decade and, subsequently, the progressive constriction of visual fields resulting in tunnel vision between the third and fifth decades of life. Progressive posterior subcapsular cataract and optic nerve head atrophy are also frequent manifestations of this *PCDH15* genotype.

*Pcdh15^{R250X}* mutant mice have a retinal dysfunction as early as 1 month after birth. We do not observe gross retinal damage, however, in aged mutant mice we do observe marginal loss of ONL thickness. Our data indicate that protocadherin-15 has a dual role in PRs and the RPE. First, at the junction of the PR IS and OS, where protocadherin-15 is localized (***Reiners et al., 2005a***), the loss of protocadherin-15 leads to disrupted shuttling of arrestin and transducin under light-adapted conditions. Second, within the RPE, loss of protocadherin-15 is associated with lower levels of two key visual retinoid cycle enzymes, CRALBP and RPE65. Reduced levels of RPE65 were reported in *Myo7a* knockout mice (***Lopes et al., 2011***), a binding partner of protocadherin-15. Further, CRALBP facilitates the transport of 11-*cis* retinal between the RPE and the PR OS (***Saari et al., 2001***). Hence, reduced levels of RPE65 and CRALBP lead to delayed and reduced regeneration and transport of 11-*cis* retinal to the PR OS, and thus we observed a concordant reduction in levels of 11-*cis* retinal oxime. Our data provide a plausible explanation for reduced ERG amplitudes without gross retinal degeneration in *Pcdh15^{R250X}* mutants, suggesting this may also be the case for other *Pcdh15* mutant mice (***Garwin et al., 2000***; ***Libby and Steel, 2001***; ***Liu et al., 2007***; ***Peng et al., 2011***).

Unlike the typical human ocular manifestations of USH1, which have severe retinal degeneration, our mouse model has much less severe pathophysiology. The discordance between retinal pathologies in humans and mice may be further attributable to the structural differences in their PRs, particularly the presence of calyceal process in humans, monkeys, and frogs, but not in rodents (***Sahly et al., 2012***), light exposure (***Lopes et al., 2011***), or environmental factors. The role of Usher proteins in the calyceal processes is supported by recent observations of PR degeneration in *Ush1* frog models that have calyceal processes (***Schietroma et al., 2017***). Further, this is consistent with reported ocular phenotypes of other USH mouse models on C57BL/6J background (***Garwin et al., 2000***; ***Jacobson et al., 2008***; ***Liu et al., 2007***; ***Liu et al., 1999***; ***Williams et al., 2009***). However, a recent study showed degeneration of cone PRs in *Ush1c* and *Ush1g* knockout mice on an albino background (***Trouillet et al., 2018***), which suggests that pigmentation might be providing protection to mice against the ambient light condition in their housing facilities. Currently, we are backcrossing *Pcdh15^{R250X}* to generate congenic mice with an albino background. Future studies will assess the PR fate and ERG progression in these mice.

In conclusion, documenting the natural history and the degree of clinical variability of the ocular phenotype in human and animal models is pivotal for evaluating the efficacy and potential therapeutics

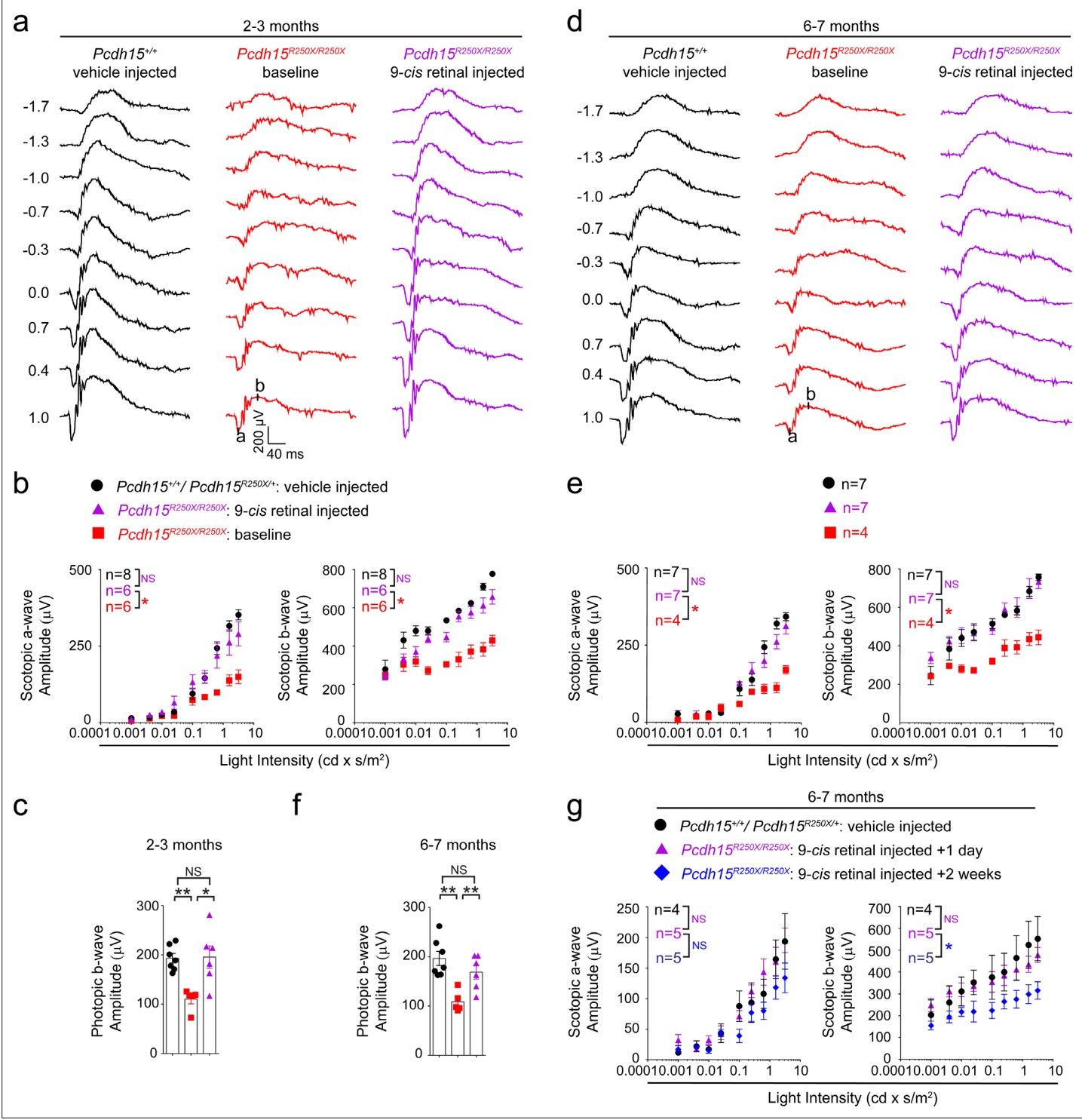

**Figure 4.** Exogenous 9-*cis* retinal rescues ERG deficits in young and old mutant mice. (**a, d**) Representative scotopic ERG traces from young (2–3 months) (**a**) and old (6–7 months) (**d**). 9-*cis* retinal injected *Pcdh15* mutant mice (*right* panels) show waveforms comparable to vehicle-injected control mice (*left* panels). Same *Pcdh15*[R250X] mutant mice assessed 1 week prior to 9-*cis* retinal injection and ERG assessment show significantly reduced waveforms (*central* panels, baseline). (**b, e**) a- (*left* panel) and b-wave (*right* panel) quantification of scotopic ERG amplitudes shown in (**a**) and (**d**), respectively. (**c, f**) Quantification of photopic b-wave for the denoted mice shows 9-*cis* retinal also improved cone-mediated function of mutant mice. (**f**) a- (*left* panel) and b-wave (*right* panel) quantification of scotopic ERG amplitudes for a different cohort of 6–7-month-old mice show that 2 weeks after 9-*cis* retinal injection in mutant mice, the efficacy starts to wane. Data presented as mean ± SEM. One-way ANOVA and Bonferroni post hoc test, $p < 0.05$

*Figure 4 continued on next page*

*Figure 4 continued*

(*) or p<0.001 (***). NS, not significant. The online version of this article includes source data and the following figure supplement(s) for Figure 4. ERG, electroretinography.

The online version of this article includes the following figure supplement(s) for figure 4:

**Figure supplement 1.** Accumulation of various retinoids in tissues following exogenous 9-*cis* retinal delivery.

**Figure supplement 2.** Exogenous 9-*cis* retinal does not improve the mislocalization of key phototransduction cascade proteins.

in future clinical trials. Our longitudinal USH1F patient ocular data show that significant vision and PRs are preserved until the third decade of life, providing a long window of opportunity. Our results with an 11-*cis* retinal analog, 9-*cis* retinal, raise the possibility that longer-lasting analogs such as 9-*cis* retinyl acetate, which has an excellent safety profile (*Koenekoop et al., 2014*; *Scholl et al., 2015*) or a synthetic version of 11-*cis* β-carotene, whose capsule formulation is already approved by the United States Food and Drug Administration (*Rotenstreich et al., 2013*), could preserve vision in USH1F patients. Furthermore, in mouse models lacking key visual cycle enzyme RPE65 or with one of the most common variants of opsin causing RP (p.Pro23His), administration of retinoids has been shown to preserve PR morphology or proper folding of opsin to a greater extent (*Maeda et al., 2009*; *Noorwez et al., 2004*), and thus might also extend the life of functional PRs in USH1 patients. Based on our pre-clinical data in mouse and prior human trials, a clinical trial in USH1F patients may show benefit if the retinoid is administered early in life.

# Materials and methods

## Key resources table

| Reagent type (species) or resource | Designation | Source or reference | Identifiers | Additional information |
|---|---|---|---|---|
| Gene (*mouse*) | *Pcdh15* | GenBank | Gene ID: 11994 | |
| Gene (*human*) | *PCDH15* | GenBank | Gene ID: 65217 | |
| Genetic reagent (*Mus musculus*) | *Pcdh15^R250X^* | This paper | | |
| Antibody | Protocadherin-15 (Rabbit polyclonal) | *Ahmed et al., 2003a* | PB303; C-terminus | IF (1:200) |
| Antibody | Opsin (Ret-P1; mouse monoclonal) | MilliporeSigma | MAB5316 | IF (1:500) |
| Antibody | Transducin (mouse monoclonal) | Santa Cruz Biotechnology | Sc-517057 | IF (1:100) |
| Antibody | Arrestin (mouse monoclonal) | Drs. Paul Hargrave and Clay Smith, University of Florida, FA | clone C10C10 | IF (1:25) |
| Antibody | Protocadherin-15 (Rabbit polyclonal) | *Ahmed et al., 2006* | HL5614; N-terminus | IF (1:1200) |
| Antibody | Transducin (mouse monoclonal) | Santa Cruz Biotechnology | Sc-517057 | IF (1:100) |
| Antibody | Rhodopsin clone RET-P1 (mouse monoclonal) | EMD Millipore | MAB5316 | IF (1:250) |
| Antibody | Actin clone 13E5 (Rabbit polyclonal) | Cell Signaling Technology | 4970 S | IF (1:200) |
| Probe | Rhodamine phalloidin | Thermo Fisher Scientific | R415 | IF (1:200) |
| Antibody | IRBP (Rabbit polyclonal) | Santa Cruz Biotechnology | Sc-25787 | WB (1:500) |

*Continued on next page*

*Continued*

| Reagent type (species) or resource | Designation | Source or reference | Identifiers | Additional information |
|---|---|---|---|---|
| Antibody | CRALBP (Rabbit polyclonal) | Santa Cruz Biotechnology | Sc-28193 | WB (1:1000) |
| Antibody | RPE65 (Rabbit polyclonal) | Kind gift from Dr. Michael Redmond, National Institutes of Health, Bethesda, MD | – | WB (1:100) |
| Chemical compound, drug | 9-*cis* retinal | Sigma-Aldrich | R5754 | |
| Commercial assay or kit | ECL Prime Western Blotting System | Thermo Fisher Scientific | 32,106 | |
| Other | FM1-43 | Thermo Fisher Scientific | T3163 | Dye |
| Other | Chaps Hydrate ≥98% (HPLC) | Sigma-Aldrich | SIG-C3023-25G | Detergent |

## Patient assessment

The records of 13 patients were reviewed under the National Eye Institute, National Institutes of Health protocol 08-EI-N014. Informed consents were obtained from the patients to conduct this research. Eleven were homozygous for the p.Arg245* founder variant was associated with the majority of USH1 of the Ashkenazi Jews in our study and two subjects who were compound heterozygous with one p.Arg245* variant and a second pathogenic variant p.Arg929* of *PCDH15* in trans. Data included demographic information, age of onset of visual symptoms, date of ophthalmic exams and reason for visit, BCVA, visual fields, presence and type of lenticular opacities, fundus exam findings, Optical Coherence Retinography when available, and electroretinography. The horizontal diameter of the V4e, I4e, and I1e isopters on Goldmann Visual Fields were measured. Five patients were seen at the NIH Clinical Center under protocol 05-EI-0096 and three had follow-up visits over a 30-year period. These patients underwent a complete ophthalmologic examination including BCVA with manifest refraction, biomicroscopy, and photography of lens opacity, if present, and visual field evaluation by Goldmann kinetic perimetry. Dilated ophthalmoscopic examination was performed after instillation of phenylephrine 2.5% and tropicamide 1%. Digital photography of the retinal fundus was performed. Snellen visual acuity was measured using ETDRS charts. In patients whose visual acuity was reduced to a degree preventing them from reading the chart, the ability to recognize hand motion (HM) or perceive light (LP) was documented. The presence or absence of cystoid macular changes and/or atrophic pigmentary macular changes were assessed by ophthalmoscopy and/or macular photography.

## Generating *Pcdh15^R250X^* mice

*Pcdh15^R250X^* mice were generated by the Cincinnati Children's Hospital Genetics Core using CRISPR/Cas9 technology and then transferred to the University of Maryland School of Medicine (UMSOM) facilities. In addition to the desired mutation (in red, see below) that changes the codon of R250 from CGA to TGA, silent mutations were introduced to create a Hae II restriction endonuclease site (underlined) which is used to facilitate genotyping as well as to prevent recutting by Cas9 nuclease.

> PAM.
> Wt: …GACCGTGCACAAAATCTGAATGAGAGGCGAACAACCACCA….
> R.
> X.
> KI: …GACCGTGCACAAAATCTGAATGAGcGctGAACAACCACCA….
> Hae II.

Heterozygous founder mice were bred with WT C57BL/6J mice and the colony was further expanded on the C57BL/6J background. Mice are genotyped using primers VS4576: TTCACCTTCCATTCCCCCAAC and VS4577: CTTACCGGAGTCCTCAGTTCAGG, which generates a 343- bp amplicon that was also Sanger sequenced. Mice were housed in a facility with 12 hr of light and 12 hr with the lights off. Mice were fed after weaning on a standard mouse diet and with water available ad libitum. We followed the ARRIVE guidelines for reporting animal research and studies were conducted in accordance with the ARVO Statement for the Use of Animals in Ophthalmic and Vision Research as

well as the National Institutes of Health Guide for the Care and Use of Laboratory Animals. All animal procedures were approved by the UMSOM IACUC (Institutional Animal Care and Use Committees).

## Electroretinography and optical coherence tomography

ERGs were recorded as previously described (*Sethna et al., 2016*). Overnight dark-adapted mice were anesthetized with a combination of ketamine-xylazine (100 mg/kg and 10 mg/kg, respectively), followed by dilation of pupils with 1% Tropicamide. A gold loop wire electrode was placed on the cornea, a reference electrode was placed on the scalp under the skin and a ground electrode was placed under the skin near the tail. ERG waveforms were acquired using sequentially brighter stimuli (0.003962233–3.147314 cd × s/m$^2$) with 5–60 s intervals using the Diagnosys ColorDome Ganzfeld system (Diagnosys Systems, Lowell, MA). Three to five waveforms per intensity were averaged. Photopic, cone-only, responses were acquired at a single bright flash (3.15 cd × s/m$^2$) under a steady rod-suppressing field of 30 cd × s/m$^2$, with 10 waves averaged. Waves were analyzed using inbuilt Espion software. For exogenous 9-*cis* retinal treatment, animals received intraperitoneal 0.25 mg 9-*cis* retinal (Sigma-Aldrich Inc, Saint Louis, MO) (25 mg dissolved in 200 µl 100 % ethanol) and diluted 1:10 in vehicle (180 µl sterile filtered 10% BSA in 0.9% NaCl solution) or vehicle-only (20 µl 100% ethanol and 180 µl 10% BSA in 0.9% NaCl solution), in the dark (*Sethna et al., 2020*; *Xue et al., 2015*). Animals were dark adapted overnight and ERGs were performed as above. OCT was performed using Spectralis OCT (Heidelberg Engineering, Heidelberg, Germany). Mice were anesthetized and dilated as above. A custom-designed plano-concave contact lens micro-M 2.00/5.00 (Cantor & Nissel Ltd, Northamptonshire, UK) was used to obtain cross sections of the entire retina. Outer nuclear quantification was performed as detailed before (*Zeng et al., 2016*).

## Immunohistochemistry (eye and ear) and FM1–43 uptake

Mice ( 1–3-month-old) were dark adapted overnight and euthanized before light onset and eyes were enucleated following CO$_2$ asphyxiation followed by cervical dislocation or exposed to normal room light for 2 hr after light onset and euthanized as above and processed as below. Dark-adapted procedures were performed under very dim red light. Eyes were immediately fixed in Prefer fixative (Anatech LTD, Battle Creek, MI), paraffin embedded and stained using standard protocols (*Sethna et al., 2016*; *Sethna and Finnemann, 2013*). Briefly, 7 µm sections were deparaffinized, rehydrated in phosphate-buffered saline (PBS), blocked and permeabilized with 10% normal goat serum/ 0.3% Triton-X 100 for 2 hr at room temperature (RT), and incubated overnight at 4°C with the indicated primary antibodies to transducin (1:100 dilution, #Sc-517057, Santa Cruz Biotechnology, Dallas, TX) and arrestin (clone C10C10, 1:25 dilution, kind gift from Drs. Paul Hargrave and Clay Smith, University of Florida, FA). The following day, sections were incubated with Alexa fluor labeled goat secondary antibodies (1:250) and DAPI (Thermo Fisher Scientific, Waltham, MA) to label nuclei. Sections were scanned using the UMSOM core facility Nikon W1 spinning disk microscope and images were processed using FIJI software (*Schindelin et al., 2012*). To stain for protocadherin-15, dissected eyes were fixed in 4% paraformaldehyde (Electron Microscopy Sciences, Hatfield, PA) and processed as above using a previously described custom antibody targeting the C-terminus of protocadherin-15 (PB303; 1:200) (*Ahmed et al., 2003a*).

P60 temporal bones were fixed and processed for immunocytochemistry as previously described (*Riazuddin et al., 2012*). The cochlear and vestibular sensory epithelia were isolated, fine dissected, and permeabilized in 0.25% Triton X-100 for 1 hr and blocked with 10% normal goat serum in PBS for 1 hr. Tissue samples were probed overnight with antibodies against myosin VIIa or custom antibody targeting the N-terminus of antibody protocadherin-15 (HL56614; 1:200 dilution)(*Ahmed et al., 2006*), and after three washes, were incubated with the secondary antibody for 45 min at RT. Rhodamine phalloidin was used at a 1:250 dilution for F-actin labeling. All images were acquired using an LSM 700 laser scanning confocal microscope (Zeiss, Germany) using a 63× 1.4 NA or 100× 1.4 NA oil immersion objectives. Stacks of confocal images were acquired with a Z-step of 0.5 µm and processed using ImageJ software (National Institutes of Health, Bethesda, MD).

Cochlear and vestibular explants were dissected at postnatal day 0 (P0) and cultured in a glass-bottom petri dish (MatTek, Ashland, MA). They were maintained in Dulbecco's modified Eagle's medium supplemented with 10 % fetal bovine serum (Thermo Fisher Scientific, Waltham, MA) for 2 days at 37°C and 5% CO2. Explants were incubated for 10 s with 3 µM FM1–43, washed three

times with Hank's balanced salt solution, and imaged live using a Zeiss LSM 700 scanning confocal microscope.

## Auditory brainstem response measurements

Hearing thresholds of heterozygous and homozygous $Pcdh15^{R250X}$ mice at P16 (n=5 each genotype) were evaluated by recording ABRs. All ABR recordings, including broadband clicks and tone-burst stimuli at three frequencies (8, 16, and 32 kHz), were performed using an auditory-evoked potential RZ6-based auditory workstation (Tucker-Davis Technologies Alachua, FL) with a high frequency transducer. Maximum sound intensity tested was 100 dB SPL. TDT system III hardware and BioSigRZ software (Tucker Davis Technology, Alachua, FL) were used for stimulus presentation and response averaging.

## Vestibular testing

Exploratory tests were performed as previously described (*Michel et al., 2017*). Briefly, mice were placed individually in a new cage. A camera was placed on top of the cage to record movements of mice for 2 min and tracked using ImageJ software. Tail hanging tests were performed as follows: mice were held 5 cm above a tabletop. The test scores were given as following: normal behavior was demonstrated by a 'reaching position,', with a score of 4, by the extension of limb and head forward and downward aiming to the tabletop. Mice with abnormal behavior, ranked with a score of 1, tried to climb towards the examiner's hand, curling the body upward reaching with the head to the tail one time. Mice ranked with a score of 0, tried to climb toward the examiner's hand, curling the body upward reaching with the head to the tail multiple times.

## Retinoid extraction and analysis

All procedures for retinoid extraction were performed under red safelights. Overnight dark-adapted mice were euthanized with $CO_2$, eyes enucleated, lens and vitreous removed, followed by freezing the eyecups in pairs on dry ice. These were stored at – 80°C until retinoid extraction was performed. Mouse eyecup pairs were homogenized in fresh hydroxylamine buffer (1 ml of 0.1 M MOPS, 0.1 M $NH_2OH$, and pH 6.5). 1 ml ethanol was added, samples were mixed and incubated (30 min in the dark at RT). Retinoids were extracted into hexane (2×4 ml), followed by solvent evaporation using a gentle stream of argon at 37°C. After reconstituting in 50 µl mobile phase, the retinoid samples were separated on LiChrospher Si-60 (5 µm) normal-phase columns (two 2.1×250 mm in series; ES Industries, West Berlin, NJ) using an H-Class Acquity UPLC (Waters Corp., Milford, MA) along with standards at a flow rate of 0.6 ml/min, following published methods (*Landers and Olson, 1988*). Retinaloxime standards were prepared from 9-*cis*-retinal (Toronto Research Chemicals, Toronto, Canada), 11-*cis*-retinal (National Eye Institute, NIH), and all-*trans*-retinal (Sigma-Aldrich, Saint Louis, MO) using published methods (*Garwin et al., 2000*). Also, synthetic retinyl palmitate (Sigma-Aldrich Inc, St. Louis, MO) was used as a standard. Absorbance was monitored at 350 nm for retinaloximes and at 325 nm for retinyl esters. Peak areas were integrated and quantified using external calibration curves. Data were analyzed using Empower three software (Waters Corp., Milford, MA).

## Data analysis

Four to eight animals per time point/genotype/treatment for ERG analysis were used. One-way ANOVA with Tukey's post hoc test or Student's t-test was used to compare the control sample to test samples with the data presented as mean ± SEM. Differences with $p<0.05$ were considered significant. Data were analyzed using GraphPad Prism (GraphPad Software, Inc, La Jolla, CA).

## Acknowledgements

The authors thank all the probands for participating in the natural history studies. The authors thank Dr. Ekaterina Tsilou for clinical assessments, Ms. Amy Turriff and Ms. Meira Meltzer for genetic counseling, Ms. Dimitria Gomes and Mr. Samuel Garmoe for technical assistance with mice and the UMSOM core facility for access to a Zeiss-710 confocal and Nikon W1 microscopes. The authors appreciate review of the manuscript by Drs. Wade Chien and Isabelle Roux. The natural history project at the National Eye Institute (NEI) and National Institute on Deafness and Other Communication Disorders (NIDCD) was supported (in part) by intramural funds to W.M.Z. and T.B.F (DC000039), respectively. Work at

the University of Maryland Baltimore was supported by Research Funds from Usher1F Collaborative Foundation award (Z.M.A.).

# Additional information

## Competing interests

Julie M Schultz: Julie M. Schultz is affiliated with GeneDx, Inc. The author has no financial interests to declare.. The other authors declare that no competing interests exist.

## Funding

| Funder | Grant reference number | Author |
|---|---|---|
| USHER 1F Collborative | Usher1F | Zubair M Ahmed |
| National Institute on Deafness and Other Communication Disorders | DC000039 | Thomas B Friedman |

The funders had no role in study design, data collection and interpretation, or the decision to submit the work for publication.

## Author contributions

Saumil Sethna, Conceptualization, Formal analysis, Investigation, Methodology, Supervision, Validation, Visualization, Writing – original draft, Writing – review and editing; Wadih M Zein, Conceptualization, Data curation, Formal analysis, Funding acquisition, Investigation, Visualization, Writing – original draft, Writing – review and editing; Sehar Riaz, Data curation, Formal analysis, Investigation, Writing – review and editing; Arnaud PJ Giese, Data curation, Formal analysis, Investigation, Visualization, Writing – review and editing; Julie M Schultz, Data curation, Formal analysis, Investigation, Validation, Writing – review and editing; Todd Duncan, Data curation, Formal analysis, Investigation, Methodology, Visualization, Writing – review and editing; Robert B Hufnagel, Carmen C Brewer, Formal analysis, Resources, Writing – review and editing; Andrew J Griffith, Conceptualization, Formal analysis, Funding acquisition, Resources, Supervision, Visualization, Writing – review and editing; T Michael Redmond, Formal analysis, Funding acquisition, Project administration, Resources, Supervision, Visualization, Writing – review and editing; Saima Riazuddin, Funding acquisition, Investigation, Project administration, Resources, Supervision, Visualization, Writing – review and editing; Thomas B Friedman, Conceptualization, Formal analysis, Funding acquisition, Project administration, Resources, Supervision, Visualization, Writing – original draft, Writing – review and editing; Zubair M Ahmed, Conceptualization, Formal analysis, Funding acquisition, Investigation, Project administration, Resources, Supervision, Validation, Writing – original draft, Writing – review and editing

## Author ORCIDs

Thomas B Friedman http://orcid.org/0000-0003-4614-6630
Zubair M Ahmed http://orcid.org/0000-0003-2914-4502

## Ethics

Human subjects: The records of 13 patients were reviewed under the National Eye Institute, National Institutes of Health protocol 08-EI-N014. Informed consents were obtained from the patients to conduct this research.
We followed the ARRIVE guidelines for reporting animal research and studies were conducted in accordance with the ARVO Statement for the Use of Animals in Ophthalmic and Vision Research as well as the National Institutes of Health Guide for the Care and Use of Laboratory Animals. All animal procedures were approved by the UMSOM IACUC (Institutional Animal Care and Use Committees).

## Decision letter and Author response

Decision letter https://doi.org/10.7554/eLife.67361.sa1
Author response https://doi.org/10.7554/eLife.67361.sa2

## Additional files

### Supplementary files
- Transparent reporting form
- Source data 1. Data for all figures and figure supplements.

### Data availability
All data generated or analyzed during this study are included in the manuscript and supporting files. Source data for all figures is provided in Source Data 1.

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
