## [Decision Letter]

**Acceptance summary:**

The study showed progressive retinal degeneration in Usher syndrome type I patients with the p.Arginine 245 variant of the PCDH15 gene. In mouse mutants with an equivalent variant there was impaired light-dependent translocation of arrestin and transducin, and reduced retinoid levels. Systemic supplementation of 9-cis retinal, improved retinal function suggesting a potential therapeutic approach.

**Decision letter after peer review:**

Thank you for submitting your article "Potential therapy for progressive vision loss due to PCDH15-associated Usher Syndrome developed in an orthologous mouse" for consideration by *eLife*. Your article has been reviewed by 3 peer reviewers, one of whom is a member of our Board of Reviewing Editors, and the evaluation has been overseen by Mone Zaidi as the Senior Editor. The following individuals involved in review of your submission have agreed to reveal their identity: Henri Leinonen (Reviewer #2); Prof. Claudio Punzo (Reviewer #3).

Essential revisions:

1. The paper nicely links clinical phenotype, disease model generation and finding a new therapeutic approach. The mouse phenotyping was done quite extensively. However, the main topic (it is even in the title) of the manuscript is the novel therapy, but then the drug research part itself is the smallest in the whole MS.

2. The level of 11-cis-retinal downregulation could explain the mouse phenotype the authors see. But the finding of arrestin and transducin mis-trafficking was a bit confusing. Does this fact affect visual function and how does the retinoid supplementation therapy address this?

3. How relevant is this mouse model translationally since there is no progression, and the human condition is progressive? Then, in humans, is progression caused by defective visual cycle or phototransduction protein mislocalization? The authors need to discuss this.

4. The paper requires elaboration of the proposed main mechanisms of visual function loss because the proposed drug treatment is specific to this mechanism. It also requires also clear presentation, or citing of previous literature, that the mutated gene/protein exists in main locus of the dysfunction, the RPE.

5. The authors find major types of mechanistic dysfunction in the retina; dysfunctional trafficking of arrestin/transducin and impaired visual cycle. Their treatment specifically addresses visual cycle, and vision is practically fully restored. Therefore, is dysfunctional trafficking of arrestin/transducin insignificant? Or is it also secondarily corrected by the retinoid supplementation?

6. A few experiments specifically to study visual cycle function are needed, because the treatment is fully dependent on this.

7. The specific experiments concerning how authors could elaborate the degree of visual cycle defect are:

a. Either dark-adaptation recovery with ERG, for example as in Figure 5 here:

https://faseb.onlinelibrary.wiley.com/doi/10.1096/fj.201902535R

b. Or retinoid recovery kinetics after strong bleach:

https://academic.oup.com/hmg/article/27/13/2225/4969374

Figure 7 here. Three time points (e.g. 1 h, 4 h and 8 h) could be sufficient. Essentially, the authors already have 0 h time point as they did dark-adapted one.

8. How was the RPE affected in patients and mice with PCDH15 mutation? And with treatment?

9. One single IP injection of 9-cis retinal was shown to preserve retinal function in mouse mutants, levels of 9-cis retinal in circulation and retinas should be included after injection and the duration of the effect noted

10. Would 9-cis retinal administration prevent mislocalization of transducin and arrestin in light-adapted mice?

11. At least 2 mice per group should be added to 1-mon old ERG data set (current n=4) to increase n up to 6.

*Reviewer #1:*

1. How was the RPE affected in patients and mice with PCDH15 mutation? And with treatment?

2. One single IP injection of 9-cis retinal was shown to preserve retinal function in mouse mutants, levels of 9-cis retinal in circulation and retinas should be included after injection and the duration of the effect noted

3. Would 9-cis retinal administration prevent mislocalization of transducin and arrestin in light-adapted mice?

4. At least 2 mice per group should be added to 1-mon old ERG dataset (current n=4) to increase n up to 6.

*Reviewer #2:*

Usher Syndrome is a rare genetic disorder characterized by deafness, vestibular (body balance) problems and progressive loss of vision. Usher syndrome is divided into 3 subtypes differing in progression and severity such that type 1 is the most dramatic. Saumil and colleagues´ manuscript concerns Usher Syndrome type 1 (USH1) and proposes a novel treatment strategy to mitigate visual dysfunction in affected patients. In their manuscript Saumil et al., first present longitudinal phenotype characterization of a patient population affected mutation in the gene encoding protocadherin-15 (PCDH15) that is one of the gene mutations causative for USH1. They then generate a novel mouse model carrying the same mutation and perform a thorough phenotype characterization including auditory, vestibular and visual system; however, primarily focusing on the eye. They find that the Pcdh15 mutant mice have compromised photoreceptoral trafficking of phototransduction proteins transducin and arrestin at the dark-light switch, and abnormally low expression of visual cycle-related proteins RPE65 and CRALBP as well as lowered level of visual chromophore in their retinas culminating into retinal function impairment. The authors then test the hypothesis if chromophore supplementation by exogenous 9-cis-retinal could mitigate retinal dysfunction, which proves to be true: a single i.p. injection of 9-cis-retinal essentially rescued visual function to WT levels. The concept bears clinical importance as chromophore supplementation therapies have been used in other retinal degenerative diseases before and have proved acceptable safety profile.

The patient phenotype follow-up is uniquely longitudinal and quite comprehensive. Establishment and characterization of the new mouse model with respect to the inner ear problems is adequate. Where the current manuscript falls short is confirmation that visual cycle defect is the main cause of visual dysfunction. The authors base their conclusion on lowered visual cycle-protein expression levels and a single time-point retinoid level analysis. This poorly establishes the breadth of visual cycle dysfunction. The authors should test the kinetics of retinoid regeneration after a strong bleach and also preferably test how dark-adaptation differs from healthy mice using electroretinography (a dark-adaptation recovery test, well characterized in literature). The authors should also clearly and discreetly explain/show the expression level of PCDH15 in the RPE, which was not clear to me. Another puzzling thing is the defect in phototransduction-protein trafficking at dark-light switch. How much does this contribute to the observed visual dysfunction, and if it does, by what mechanism a single administration 9-cis-retinal corrects this defect as well, or does it? Authors should test transducin and arrestin trafficking during/after their treatment, similarly as they did in mouse phenotyping part. Finally, the authors did not quantitatively test presence of retinal degeneration in the mouse model. OCT was done only at 1-month of age and statistical analysis of morphometrical parameters is not shown. Nevertheless, due to its clinical applicability retinoid supplementation should be tested in PCDH15-mutation affected USH1 individuals as only a single dose could prove its potential.

– I suggest to use presence tense "reveals" instead of "revealed" in the manuscript main title.

– Transducin not transducing in the abstract.

– What are the retinoid products FDA-accepted for clinical use in retinal degeneration?

– In introduction, general prevalence of the condition is missing. How many patients per million inhabitants? This may not be known, but at least some kind of estimate would be good to disclose for readers.

– In row 81 authors write about vision preservation. In reality, the PCDH15 mutation causes also other problems than visual cycle, as authors even show themselves. How does the retinoid therapy tackle these issues? Maybe safer to say "improve visual function".

– In row 91, the authors talk about ERG defects and refer to table 1, but I do not see any parameters of ERGs in the table.

– Row 140: I suggest to remove "outer nuclear" from bipolar neurons. Kind of confusing since BCs are located in the inner nuclear layer.

– Supplementary figure 3. Quantification of retinal layers from OCT images is missing. Best if authors did this in the older mice they tested with ERG. Based on the arrestin/transducin trafficking and visual cycle issues, one would expect there to be at least a slowly progressive degeneration (which may not be so readily detectable from current ERG follow-up).

– Major comment: visual cycle defect needs to be better characterized by a bleaching challenge test. Please perform a strong bleach and chromophore regeneration assessment preferably coupled with dark-adaptation test with ERG.

– Row 203: Current journal policies in general do not like (data not shown).

– Did the authors produce the 9-cis-retinal or where was it obtained from?

– Row 324-325: ethanol is highly toxic in mice. What was the total ethanol amount injected into mice?

– Retinoid extraction and analysis: Details missing how long the mice were dark-adapted.

– Data analysis: why/how did the authors use one-way ANOVA when there are two factors: treatment group as a between-subjects factor and flash intensity as a within-subjects factor. Would two-way ANOVA be more appropriate? Were the pre-assumptions tested and met for parametric tests, such as normality of the data?

– Data presentation in general. Largest amount of data granularity is encouraged, in keeping with the clarity of presentation. For example, in Figure S2C would be nice to see replicates. The number of biological/technical replicates need to be apparent in the M%M section and in figure legends.

– In Figure 1d, it would be nice to see the disease stage (e.g. age) associated with each of the panels in the figure (not figure legend).

– Figure 2: Do the authors mean log cd´s/m2 in the x-axis?

– Figure 2 legend says that t-test is applied. This is not proper for the ERG data performed with several flash intensities. Change of type 1 error becomes very prevalent, due to multiple comparisons.

Figure 2. Could the authors comment about the retinal sensitivity. It seems that sensitivity (I50) is not declined, but rather Rmax. Would significant visual cycle defect cause retinal drop in sensitivity (I50) parameter?

*Reviewer #3:*

The manuscript by Sethna et al., decribes a new mouse model based on a founder mutation in the Usher syndrome type 1 gene called Protocadherin-15, known to cause Usher syndrome type 1F. The mouse model recapitulates most aspects of the human disease including hearing loss and sever visual dysfunction. However, unlike in humans there seems to be no apparent retinal degeneration. Unfortunately, this is a common problem seen with many gene mutations known to cause Usher syndrome. The authors therefore characterize the mouse model further with the aim to determine what underlies the visual dysfunction. They determine that there are problems in the visual cycle in these mice. In particular, the translocation of arrestin and transducin in light adapted mice is wrong as they are suddenly present throughout the inner and outer segments. They also identify reduced CRALBP and RPE65 levels in these mice. Both these proteins play an intricate role in the recycling of the visual chromophore. Consequently these mice had also reduced levels of 11-cis retinal. Based on their findings and previous reports in the literature they treated their mice with 9-cis retinal, which is a analog of 11-cis retinal that has more stable chemical properties and is easier to synthesize and administer. Treatment of mice with 9-cis retinal significantly improved visual function, even in 6 months old mice. The authors conclude that patients suffering from this particular founder mutation in Protocadherin-15 could be potentially treated with 9-cis retinal in a clinical trial. In particular, since the compound is already FDA approved.

The manuscript is well written and clear. major strength of the manuscript are the phenotypic analyses of 13 human patients, some over a prolonged period of time, with the same mutation as is mimicked in mouse. The recapitulation of the human phenotype in mouse and the discovery of a potential therapy in mouse add further strength to the rationale for a clinical trial in humans with the same therapeutic approach used in mouse. The data support this conclusion and idea, especially since the drug is already FDA approved. The only weakness is that the characterization of the eye phenotype in mouse is not extended beyond 6-7 months of age. While this does not diminish the conclusion it would have been interesting to see if there are phenotypes that take more time to develop in mouse. The explanation for why the authors believe there will likely be no phenotype is sound and supported by data from other Usher mouse models in the literature.

Overall this is an important study that may not only help individuals affected by this particular mutation in Protocadherin-15, but may also pave the way for similar approaches in other Usher syndrome cases. The study is therefore of broad impact.

The study by by Sethna et al., is a well developed study with a linear line that tracks through the manuscript. It is very logical and clear. There are no concerns with the study in its current form.

As mentioned in the above comments it would have been interesting to analyze the eyes also at a later time point (~1 year of age) to see if any degeneration develops slowly. While the rationale for why this is likely not the case is sound (absence of calyceal processes), given that this is a new mouse model it would have added more to the story. In case there would be some slow degeneration after a year this could have opened the opportunity to test how well the 9-cis retinal treatment works when the tissue is degenerating. After all, in humans treatment is likely start in patients that have already some signs of degeneration. Nonetheless, absence of this experiment does not diminish the conclusions nor the impact of the study.

---

## [Author Response]

Essential revisions:1. The paper nicely links clinical phenotype, disease model generation and finding a new therapeutic approach. The mouse phenotyping was done quite extensively. However, the main topic (it is even in the title) of the manuscript is the novel therapy, but then the drug research part itself is the smallest in the whole manuscript.

Thanks. We agree and provided additional data on evaluating the impact of 9-cis retinal delivery on the visual function as well as on arrestin and transducin transport.

2. The level of 11-cis-retinal downregulation could explain the mouse phenotype the authors see. But the finding of arrestin and transducin mis-trafficking was a bit confusing. Does this fact affect visual function and how does the retinoid supplementation therapy address this?

To address reviewer’s comment, we performed additional experiments, including analyses of arrestin and transducin trafficking after 9-*cis* retinal injections. We also evaluated the levels of 9-*cis* retinal in the eye and liver tissues of injected mice. Based on the findings from these experiments, we added the following sentences in the Results section:

“To confirm the 9-cis retinal delivery and metabolism, in a separate cohort of mice, 24 hours post-injection followed by 2-hour light exposure, we evaluated the retinaloxime levels, including 9-*cis* retinaloxime, both in the liver and retina (Figure S7). As expected, we found no or trace levels of 9-*cis* retinaloxime levels in the same retinae.”

“Next, we assessed whether exogenous 9-*cis* retinal treatment also improved translocation of arrestin and transducin in *Pcdh15^R250X^* mutant mice. For these studies, we injected a cohort of mutant mice with 9-*cis* retinal and control mice with vehicle. Overnight dark-adapted mice were exposed to normal room light for two hours and their retinae examined for localization of arrestin and transducin. We found mis-localization of arrestin and transducin in mutant mice injected with 9-*cis* retinal as well (Figure S8). Taken together, our data link the visual deficits to the retinoid cycle dysfunction in *Pcdh15^R250X^* mutant mice and provides a starting point to investigate the possibility of therapeutically boosting visual function in USH1F patients.”

3. How relevant is this mouse model translationally since there is no progression, and the human condition is progressive? Then, in humans, is progression caused by defective visual cycle or phototransduction protein mislocalization? The authors need to discuss this.

We agree with the reviewer that it’s important to explore the status of the visual cycle proteins in human tissue. However, USH1 subjects have a normal life span. Hence, obtaining ocular tissue in which the retina is still preserved is a limitation to correlating our observations in mouse models implicating the RPE and PR dual dysfunction. However, we did contact several EyeBanks in the USA. None of them have cadaver eye tissues from USH1F subjects. Perhaps a pig model of USH1F can be developed similar to the USH1C pig model (bioRxiv 2021.05.31.446123;), which seems to more faithfully mirror the human retinal degeneration, would be an alternative path forward.

To address the point raised by the reviewer, we have added the following sentences in the revised discussion:

“Unlike the typical human ocular manifestations of USH1, which have severe retinal degeneration, our mouse model has much less severe pathophysiology. The discordance between retinal pathologies in humans and mice may be further attributable to the structural differences in their photoreceptors, particularly the presence of calyceal process in humans, monkeys, and frog, but not in rodents (Sahly *et al.,* 2012), light exposure (Lopes *et al.*, 2011) or environmental factors. The role of Usher proteins in the calyceal processes is supported by recent observations of photoreceptor degeneration in *Ush1* frog models that have calyceal processes (Schietroma *et al.,* 2017). Further, this is consistent with reported ocular phenotypes of other USH mouse models on C57BL/6J background (Haywood-Watson *et al.*, 2006a; Jacobson *et al.*, 2008; Liu *et al.*, 2007; Liu *et al.,* 1999; Williams *et al.,* 2009). However, a recent study showed degeneration of cone photoreceptors in *Ush1c* and *Ush1g* knockout mice on an albino background (Trouillet *et al.,* 2018), which suggest that pigmentation might be providing protection to mice against ambient light condition in their housing facilities. Currently, we are backcrossing *Pcdh15^R250X^* to generate congenic mice with an albino background. Future studies will assess the photoreceptor fate and ERG progression in these mice.”

4. The paper requires elaboration of the proposed main mechanisms of visual function loss because the proposed drug treatment is specific to this mechanism. It also requires also clear presentation, or citing of previous literature, that the mutated gene/protein exists in main locus of the dysfunction, the RPE.

We agree and have added the following information in the revised manuscript:

“Immunostaining showed that protocadherin-15 is localized to the inner segments of the PR, the outer plexiform layer and the ganglion cell layer as reported previously (Haywood-Watson, Ahmed et al., 2006a). However, the RPE was not assessed for the localization of protocadherin in those studies. Here we show that protocadherin-15 is also expressed in the RPE using immunohistochemistry (Figure S2b).”

Furthermore, our new data reveals an improvement of ERG amplitudes with retinoids, but not a correction of arrestin and transducin trafficking after 9-cis retinal injection, further supporting the notion that the visual cycle deficit observed in *Pcdh15^R250X^* mutant mice stem from a retinoid cycle dysfunction.

5. The authors find major types of mechanistic dysfunction in the retina; dysfunctional trafficking of arrestin/transducin and impaired visual cycle. Their treatment specifically addresses visual cycle, and vision is practically fully restored. Therefore, is dysfunctional trafficking of arrestin/transducin insignificant? Or is it also secondarily corrected by the retinoid supplementation?

Please see our response to comment 2.

6. A few experiments specifically to study visual cycle function are needed, because the treatment is fully dependent on this.7. The specific experiments concerning how authors could elaborate the degree of visual cycle defect are:a. Either dark-adaptation recovery with ERG, for example as in Figure 5 here:https://faseb.onlinelibrary.wiley.com/doi/10.1096/fj.201902535Rb. Or retinoid recovery kinetics after strong bleach:

https://academic.oup.com/hmg/article/27/13/2225/4969374

Figure 7 here. Three time points (e.g. 1 h, 4 h and 8 h) could be sufficient. Essentially, the authors already have 0 h time point as they did dark-adapted one.

Thank you for suggesting these experiments. We have performed the experiments suggested in 7a and b and incorporated the results in Figure 3d-e and Figure S5c-d.

“Next, we quantified the retinoids levels one hour after dark adaptation following bleaching with 15,000 lux for one hour (Li *et al.,* 2019). Reduction of 11-*cis* retinaloxime in control and mutant mice retinae (Figure 3d) correlated with prebleach levels, as did increase in all-*trans* retinyl esters. These findings from *Pcdh15^R250X^* mutants suggest a reduced function of the visual cycle due to reduced expression of RPE65 and CRALBP. Since we observed lower visual cycle proteins (RPE65 and CRALBP), we also assessed the structure of the RPE, the main cell type harboring the key enzymes of the visual cycle. Transmission electron microscopy showed no gross structural deficits in the RPE (Figure S5e). Together, our data indicates that the loss of protocadherin-15 in the retina leads to aberrant translocation of proteins involved in the phototransduction cascade and reduced levels of key retinoids and enzymes involved in the visual retinoid cycle.”

8. How was the RPE affected in patients and mice with PCDH15 mutation? And with treatment?

Clinically, it is difficult to address RPE involvement separately from the photoreceptor degeneration in PCDH15 patients. We did observe macular hypoautofluorescence indicating RPE atrophy, as well as RPE and outer retinal atrophy on Optical Coherence Tomography, in LMG197#1831 starting at 38 years of age. A supplement figure (Figure S1) was added showing the left eye fundus autofluorescence and OCT findings for this patient at 50 yrs. of age. The fundus autofluorescence shows a central reduced / absent autofluorescence consistent with RPE atrophy. The OCT shows the corresponding macular area with complete RPE and outer retinal atrophy as indicated by homogenous choroidal hypertransmission and absence of RPE band.

We also performed transmission electron microscopy in mutant and control mice, which showed no gross structural deficits of the RPE (Figure S5e).

9. One single IP injection of 9-cis retinal was shown to preserve retinal function in mouse mutants, levels of 9-cis retinal in circulation and retinas should be included after injection and the duration of the effect noted

We have included the 9-*cis* retinaloxime analysis data in the revised supplementary Figure S7.

10. Would 9-cis retinal administration prevent mislocalization of transducin and arrestin in light-adapted mice?

We performed an arrestin/ transducin localization experiment under light adapted conditions after 9-*cis* retinal injections and found no correction in the localization of either proteins (Figure S8).

11. At least 2 mice per group should be added to 1-mon old ERG data set (current n=4) to increase n up to 6.

We agree and have amended the figure to now reflect an *n* of 10 mice per genotype. Please see the revised Figure 2.

Reviewer #1:1. How was the RPE affected in patients and mice with PCDH15 mutation? And with treatment?

Please refer to response 3 for human samples. We performed transmission electron microscopy in mutant and control mice, which showed no gross structural deficits of the RPE (Figure S5e).

2. One single IP injection of 9-cis retinal was shown to preserve retinal function in mouse mutants, levels of 9-cis retinal in circulation and retinas should be included after injection and the duration of the effect noted

We have included the 9-*cis* retinaloxime analysis data in the revised supplementary Figure S7.

3. Would 9-cis retinal administration prevent mislocalization of transducin and arrestin in light-adapted mice?

Our new data reveals an ERG improvement with retinoids, but not a correction of arrestin and transducin trafficking after 9-*cis* retinal injection, further support the notion that the visual cycle deficit observed in *Pcdh15^R250X^* mutant mice stem from a retinoid cycle dysfunction.

4. At least 2 mice per group should be added to 1-mon old ERG dataset (current n=4) to increase n up to 6.

We have amended the figure to reflect an *n of 10* mice per genotype. Please see the updated Figure 2.

Reviewer #2:[…]– I suggest to use presence tense "reveals" instead of "revealed" in the manuscript main title.

Fixed

– Transducin not transducing in the abstract.

Thank you for pointing this out. During submission of the abstract online, we missed the error incorporated by the spell checker resulting in “transducing”.

– What are the retinoid products FDA-accepted for clinical use in retinal degeneration?

In the revised discussion, we have added the following sentences to list current FDA-accepted retinoid products that are currently in clinical trials:

“Our results with an 11-*cis* retinal analog, 9-*cis* retinal, in a mouse model of USH1F raises the possibility that a longer lasting analog such as 9-*cis* retinyl acetate, which has an excellent safety profile (Koenekoop *et al.,* 2014; Scholl *et al.,* 2015) or a capsule formulation of a synthetic version of 11-*cis* β-carotene, which is already approved by the United States Food and Drug Administration (Rotenstreich *et al.,* 2013), may preserve vision in USH1F Usher syndrome patients.”

– In introduction, general prevalence of the condition is missing. How many patients per million inhabitants? This may not be known, but at least some kind of estimate would be good to disclose for readers.

We have added the following sentences in the introduction:

“Usher syndrome (USH) is estimated to be responsible for more than 50% of deaf-blind cases, 8-33% of patients with RP and 3-6% of congenitally deaf individuals (Boughman *et al.,* 1983; Brownstein *et al.,* 2004b; Vernon, 1969). Clinical data review studies estimated a prevalence of 3.2 to 6.2 per 100,000 for USH cases (Boughman and Fishman, 1983; Koenekoop *et al.,* 1993). However, a molecular diagnosis study in children with hearing loss found variants in USH-associated genes in 11% and estimated a frequency of 1/6000 individuals afflicted with USH in the USA (Kimberling *et al.,* 2010). Assuming similar prevalence, this would translate into 255,000 to 1.34 million USH cases worldwide. However, this estimate varies considerably in specific population substructures. For instance, the p.Arg245* founder variant of *PCDH15* (*USH1F*) has ~2% carrier frequency amongst Ashkenazi Jews accounts for nearly 60% of their USH1 cases (Ben-Yosef *et al.,* 2003). Thus, we speculate that a comprehensive understanding of the pathophysiology and disease mechanisms is a prelude for developing therapeutic interventions for Usher syndrome after clinical trials.”

– In row 81 authors write about vision preservation. In reality, the PCDH15 mutation causes also other problems than visual cycle, as authors even show themselves. How does the retinoid therapy tackle these issues? Maybe safer to say "improve visual function".

Fixed.

– In row 91, the authors talk about ERG defects and refer to table 1, but I do not see any parameters of ERGs in the table.

We have fixed the error. Electroretinography recordings were at noise-level for both scotopic and photopic responses so no data to add in Table 1. Also, we have added a representative example in supplementary Figure S1**.**

– Row 140: I suggest to remove "outer nuclear" from bipolar neurons. Kind of confusing since BCs are located in the inner nuclear layer.

Fixed.

– Supplementary figure 3. Quantification of retinal layers from OCT images is missing. Best if authors did this in the older mice they tested with ERG. Based on the arrestin/transducin trafficking and visual cycle issues, one would expect there to be at least a slowly progressive degeneration (which may not be so readily detectable from current ERG follow-up).

We have quantified OCT images and added the data to Figure 2i.

– Major comment: visual cycle defect needs to be better characterized by a bleaching challenge test. Please perform a strong bleach and chromophore regeneration assessment preferably coupled with dark-adaptation test with ERG.

We have performed the bleaching challenge test and added the results in (Figure S5). We observed equivalent recovery of a-wave regardless of genotype (Figure S5d), but we did observe that initial a-wave amplitude was lower in mutant mice (Figure S5e).

– Row 203: Current journal policies in general do not like (data not shown).

Fixed.

– Did the authors produce the 9-cis-retinal or where was it obtained from?

Sigma Aldridge (catalog #R5754). Added this information to the Methods section.

– Row 324-325: ethanol is highly toxic in mice. What was the total ethanol amount injected into mice?

We have added details in the methods section:

“For exogenous 9-*cis* retinal treatment, animals received intraperitoneal 0.25 mg 9-*cis* retinal (Sigma Aldrich Inc, Saint Louis, MO) (25 mg dissolved in 200 µl 100% ethanol) and diluted 1:10 in vehicle (180 µl sterile filtered 10% BSA in 0.9% NaCl solution) or vehicle only (20 µl 100% ethanol and 180 µl 10% BSA in 0.9% NaCl solution), in the dark (Sethna *et al.*, 2020; Xue *et al*)”.

– Retinoid extraction and analysis: Details missing how long the mice were dark-adapted.

Overnight, which we added to the methods section.

– Data analysis: why/how did the authors use one-way ANOVA when there are two factors: treatment group as a between-subjects factor and flash intensity as a within-subjects factor. Would two-way ANOVA be more appropriate? Were the pre-assumptions tested and met for parametric tests, such as normality of the data?

We are comparing amplitudes between groups at a single intensity. We performed this analysis for each intensity, similar to previous studies (e.g. PMID: 33677964; PMID: 30018116).

– Data presentation in general. Largest amount of data granularity is encouraged, in keeping with the clarity of presentation. For example, in Figure S2C would be nice to see replicates. The number of biological/technical replicates need to be apparent in the M%M section and in figure legends.

Fixed.

– In Figure 1d, it would be nice to see the disease stage (e.g. age) associated with each of the panels in the figure (not figure legend).

Done. We have revised the figure and included ages in the panel.

– Figure 2: Do the authors mean log cd´s/m2 in the x-axis?

Yes.

– Figure 2 legend says that t-test is applied. This is not proper for the ERG data performed with several flash intensities. Change of type 1 error becomes very prevalent, due to multiple comparisons.

We analyzed amplitude at each flash intensity individually between genotype, therefore t-test is appropriate, similar to previous studies (e.g. PMID: 33677964; PMID: 30018116). However, to avoid the confusion that significance is measured through multiple comparisons, we have updated the figure and legend to reflect that significance levels are assessed by one-to-one comparison of the intensity across mutants vs wild type controls, at each intensity, through a t-test.

Figure 2. Could the authors comment about the retinal sensitivity. It seems that sensitivity (I50) is not declined, but rather Rmax. Would significant visual cycle defect cause retinal drop in sensitivity (I50) parameter?

That is a very interesting and important comment. We think that the paradox is explained by the complex phenotype of the *Pcdh15* mutant, on the one hand there is reduced expression of visual cycle proteins (RPE65 and CRALBP), and on the other hand an effect on photoreceptor biochemistry/physiology. The visual cycle in mutants, though reduced in output, is still sufficient to supply what chromophore is needed by the mutant photoreceptors so sensitivity (“I50”) is not that impacted. The reduced a-wave response (“Rmax”) observed is probably due to the impacted photoreceptor biochemistry/physiology (as illustrated by arrestin/transducin effect) while the actual threshold is not that affected.

Reviewer #3:As mentioned in the above comments it would have been interesting to analyze the eyes also at a later time point (~1 year of age) to see if any degeneration develops slowly. While the rationale for why this is likely not the case is sound (absence of calyceal processes), given that this is a new mouse model it would have added more to the story. In case there would be some slow degeneration after a year this could have opened the opportunity to test how well the 9-cis retinal treatment works when the tissue is degenerating. After all, in humans treatment is likely start in patients that have already some signs of degeneration. Nonetheless, absence of this experiment does not diminish the conclusions nor the impact of the study.

We performed additional experiments on 12 to14 month old animals and have incorporated these data in Figure 2g, h, j.